# Quantifying non-stabilizerness via information scrambling

**Arash Ahmadi\* and Eliska Greplova**

Kavli Institute of Nanoscience, Delft University of Technology, Delft, the Netherlands

\* a.ahmadi-1@tudelft.nl

## Abstract

The advent of quantum technologies brought forward much attention to the theoretical characterization of the computational resources they provide. A method to quantify quantum resources is to use a class of functions called magic monotones and stabilizer entropies, which are, however, notoriously hard and impractical to evaluate for large system sizes. In recent studies, a fundamental connection between information scrambling, the magic monotone mana and 2-Renyi stabilizer entropy was established. This connection simplified magic monotone calculation, but this class of methods still suffers from exponential scaling with respect to the number of qubits. In this work, we establish a way to sample an out-of-time-order correlator that approximates magic monotones and 2-Renyi stabilizer entropy. We numerically show the relation of these sampled correlators to different non-stabilizerness measures for both qubit and qutrit systems and provide an analytical relation to 2-Renyi stabilizer entropy. Furthermore, we put forward and simulate a protocol to measure the monotonic behaviour of magic for the time evolution of local Hamiltonians.

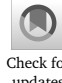

# 1   Introduction

The field of quantum computing introduced the concept that quantum systems can deliver a significant computational speed-up in a variety of settings [1–6]. Yet, although increasingly large quantum processors are available, the question remains of how to rigorously quantify the computational resources of a quantum computer. One successful approach towards determining quantum resources of a quantum state is to calculate how "far away" the state is from being possible to simulate efficiently with a classical computer [7].

A specific example of quantum states that are tractable to represent and simulate on a classical computer are the so-called stabilizer states [8]. These states result from quantum circuits produced by Clifford gates which are elements of the Clifford group generated by the Hadamard gate, the phase gate and the entangling control-NOT gate [9]. In order to get any quantum advantage over classical computers, we need to add additional gates outside of the Clifford group. By injecting more non-Clifford gates into a quantum circuit, we obtain a quantum state with further distance from a stabilizer state. This distance is in literature referred to as magic or non-stabilizerness [10]. The states that are not stabilizer states are called *magic states*. Interestingly, the Clifford operations could be easier both at the experimental level and for quantum error correction  [11–13], while universal gate-sets are achieved by the distillation of a large number of noisy magic states into a less-noisy magic state which subsequently provides the computational resources for the fault-tolerant quantum computation [7, 14–17]

Examples of magic monotones include magical cross-entropy, mana [10], and robustness of magic [15]. These measures are, however, computationally expensive to evaluate and their calculation requires exact knowledge of the wave-function combined with complex optimization [10], which excludes the study of large quantum circuits. More recently introduced magic monotonotes such as the Gottesman-Kitaev-Preskill magic measure and the stablizer Renyi entropy, [18,19], offer simplified scaling which enables exact calculation of magic for a few qubits using conventional computers.

Different approaches to describe how far a quantum state is from the stabilizer states, can also be related to the amount of quantum correlations in the system. The out-of-time ordered correlators (OTOCs) quantify quantum information scrambling [20–25]. Quantum

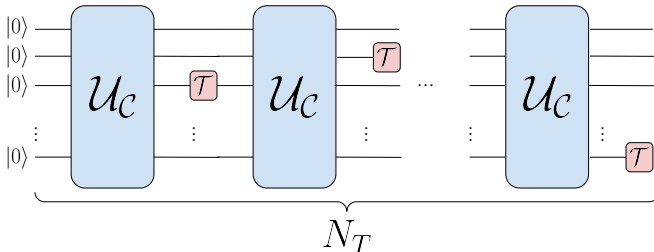

Figure 1: The schematic structure of a t-doped quantum circuit. We are using a block of the random Clifford gates, $\mathcal{U}_C$ followed by a T-gate on a random qudit. We repeat this process $N_T$ times.

information scrambling describes the spread of the local information in a quantum system [24]. Through the time evolution of a closed quantum system, the information about initial state of the system can become very hard to access due to quantum correlations in the system [26]. Even though the information is still encoded in the system it is not directly accessible without measuring all its degrees of freedom. Information scrambling has recently attracted an increasing amount of attention due to the relation with the anti-de Sitter/conformal field theory (AdS/CFT) correspondence [27]. The AdS/CFT correspondence draws a duality that relates the noise in quantum error correction codes to information scrambling in black holes [20, 22, 23]. Another application of this concept emerged in condensed matter physics such as many-body localization [28] and non-Fermi liquid behaviours [29].

Moreover, it was recently experimentally demonstrated that OTOCs can be used as an indicator of the degree of non-stabilizerness of scrambled quantum circuits [30]. In parallel, recent work has shown an analytical relation between the non-stabilizerness and OTOC [19, 31].

In this work, we show the relation between a randomised sampling of OTOC fluctuations and mana for qutrit systems and stabilizer Renyi entropy for qubit systems. We show numerical evidence that this method requires dramatically lesser number of OTOC measurements in comparison to the exact methods of calculating magic monotones. Capitalizing on this relation, we put forward an experimentally feasible way to approximate magic using the evaluation of OTOCs. Our work might lay the foundation to approximate magic in a scalable way in larger systems, as our protocol is designed to be adaptable for both numerical techniques such as tensor networks [32] and neural networks [33] as well as experimental measurements [30].

## 2 Methods

### 2.1 Magic

The concept of magic in quantum information science arises from the field of resource theory [34]. The Gottesman-Knill theorem [8] guarantees that the subset of the physical states known as *stabilizer* states are efficiently simulatable on a classical computer. More precisely, the stabilizer states are the second level of the Clifford hierarchy [9].

Since the first level (the Pauli gates) and the second level, (the Clifford gates) of the Clifford hierarchy are insufficient for universal quantum computing, we need to use the third-level gates. This level of Clifford's hierarchy includes, for example, a T-gate. Another set of important non-Clifford gates are the rotation gates $\{R_x(\theta), R_y(\theta), R_z(\theta)\}$, where $\theta$ is the angle of rotation. These gates are particularly important in problems that require a continuous set of parameters to tune, i.e. quantum machine learning algorithms [5, 6].

The amount of non-stabilizerness, or magic, of any state is measured using *magic monotones*. Magic monotones such as the robustness of magic [10] are based on an optimization over all stabilizer states, which make them practically hard to compute. However, one example of a magic monotone that does not require any optimization is known as mana, $\mathcal{M}$ [10]. This magic monotone has another limitation, namely that it is only definable for odd-prime dimensional Hilbert spaces. Additionally, mana is practically very hard to calculate since it is based on calculating discrete Wigner functions which in practice limits current calculations to at most 6 qudits. More details regarding the definition and evaluation of mana are available in Appendix A.

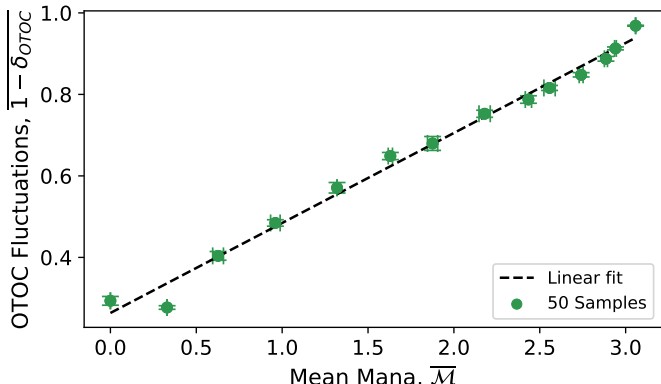

Figure 2: The fluctuation of OTOC, $\overline{1-\delta_{OTOC}}$ as a function of the mean value of mana, $\overline{\mathcal{M}}$. We see a linear behaviour between these two magic monotones for 6 qutrit t-doped circuits. Here we show results for 50 (green dots) random samples of OTOC on the y-axis. On the x-axis, we calculated mana for 10 of the samples and we fit a linear dependence (dashed line). The vertical error bar is the statistical error calculated by repeating the process above 10 more times to get the error by the standard deviation of the sampled $\delta_{OTOC}$ instances and the horizontal error bar corresponds to the standard deviation of mana.

Another method introduced to measure magic for qubits is the Stabilizer Renyi Entropy [19]. For a system of $N$ qubits, the Stabilizer Renyi Entropy of order $n$ is defined as

$$M_n(|\Psi_N\rangle) = (1-n)^{-1} \log \sum_{P \in \mathcal{P}_N} \frac{\langle \Psi_N | P | \Psi_N \rangle^{2n}}{2^N}, \tag{1}$$

where $\mathcal{P}_N$ is the set of all $N$-qubit Pauli strings and the number of the Pauli strings in $\mathcal{P}_N$ we are summing over scales as $4^N$.

## 2.2 Information scrambling

A well-known measure of information scrambling is the out-of-time-order correlators (OTOCs) which are commonly used in high-energy physics and condensed matter physics [20–25, 28, 29]. OTOC is evaluated for any two operators $A$ and $B$, where $[A, B] = 0$, as

$$\text{OTOC}(t) = \text{Re}(\langle A^\dagger(t) B^\dagger A(t) B \rangle), \tag{2}$$

where

$$A(t) = U^\dagger(t) A(0) U(t), \tag{3}$$

or equivalently

$$\text{OTOC}(U) = \frac{1}{d} tr(U^\dagger A(0) U B U^\dagger A(0) U B), \tag{4}$$

and $U(t)$ is the time evolution operator, which could either result from the time evolution of a Hamiltonian or from a quantum circuit. Here we will consider a $N$ qudit system, $A(0) = X_{N-1}$ and $B = Z_1$ where $X_i$ and $Z_i$ are the conventional Pauli operators and the subscript indicates the $i$-th qudit. As long as the commutation relation above holds, these Pauli operators can be placed on arbitrary qubit pairs. In this case, $A(0)$ plays the role of the butterfly operator related to chaotic quantum systems. The reason for using the butterfly operator is that by including a small perturbation (in this case a bit flip) we are disturbing the reversibility of the

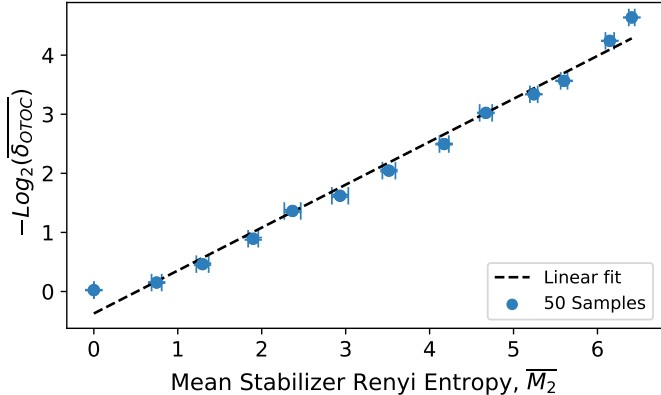

Figure 3: The log of fluctuation of OTOC, $-Log_2\overline{\delta_{OTOC}}$ over 50 (blue dots) samples as a function of the mean value of stabilizer Renyi entropy (dashed line), $\overline{M_2}$ over 10 samples. The vertical error bar is the statistical error calculated by repeating the process above 10 more times to get the error by the standard deviation of the sampled $\delta_{OTOC}$ instances, and the horizontal error bar corresponds to the standard deviation of the stabilizer Renyi entropy $M_2$.

system, which is a signature of chaos [35]. The information scrambling measured through OTOC describes how information spreads in the system and becomes inaccessible in later times [20–25]. Information scrambling also describes how local Heisenberg operators grow in time [30, 36, 37]. A way to assess how the OTOC value fluctuates over a set of random circuits is the OTOC fluctuation, $\delta_{OTOC}$, defined as the standard deviation of OTOC over all measured instances of OTOC. Let $U = C_1 V C_2$, where $V$ is a generic unitary operator. Defining the average over $C_1, C_2$ as

$$\mathbb{E}_C \text{OTOC}(U) := \int dC_1 dC_2 \text{OTOC}(U),\tag{5}$$

and define the fluctuations around the average

$$\delta_{\text{OTOC}}(U) := \mathbb{E}_C \text{OTOC}^2(U) - [\mathbb{E}_C \text{OTOC}(U)]^2.\tag{6}$$

## 3 Results

We will now numerically investigate the relation between the fluctuations of OTOC, which was experimentally observed in Ref. [30] to decrease with the growing non-stabilizerness of the quantum circuit, and the measure for magic, mana, $\mathcal{M}$ for $q = 3$ and the stabilizer Renyi entropy, $M_2$ for $q = 2$ where $q$ is the dimension of the local Hilbert space. To this end, we design random quantum circuits with Clifford and non-Clifford gates, known as t-doped quantum circuits.

### 3.1 Mana and OTOC

First, we consider $N$ qudits in $q$-dimensional Hilbert space where $q = 3$. The circuits consist of $M$ cycles of Clifford gates. In each cycle, we first apply one single Clifford gate, randomly chosen from the set $\mathcal{S} = \{H, S, X, Y, Z, I\}$ on each qudit. Then we add two CSUM gates on two randomly chosen qudits, where the CSUM gate is the counterpart of CNOT in Hilbert spaces with $q > 2$. Here we have a fixed number of $M = 10$ random cycles for each block of

random Cliffords. Finally, we add a single non-Clifford gate, $T$, on a randomly chosen qudit. We increase the magic in the circuit by increasing the number of layers of the random Cliffords followed by a T-gate.

We begin by analyzing the relationship of mana and OTOC in the Hilbert space of dimension $q = 3$ for circuits containing four qutrits such that mana is well-defined and computationally tractable. We use the qutrit Clifford gates introduced in [38]. We provide detailed definitions of all gates in Appendix B.

We observe an increasing monotonous relation between the mean value of mana, $\overline{\mathcal{M}}$ and the OTOC fluctuations, $1-\delta_{OTOC}$, see Fig. 2. In Fig. 2, we observe a linear dependence between $\overline{1-\delta_{OTOC}}$ and $\overline{\mathcal{M}}$. This relationship corresponds to the linear fit $\overline{1-\delta_{OTOC}} \approx 0.22\overline{\mathcal{M}} + 0.26$. We simulated the OTOC instances of 50 circuit runs and the number of T-gates in the circuit is $N_T \in [0, 20]$. For the simulation of the quantum circuits, we have used the Cirq package [39].

## 3.2 The Stabilizer Renyi entropy and OTOC

Mana, discussed in the previous section, is not only challenging from the scaling point of view but also only defined for odd-dimensional local Hilbert space; because it is related to the negativity of discrete Wigner functions, and thus not possible to evaluate for qubits [40, 41]. In this section, we investigate the relation of 4-OTOC fluctuations, $\delta_{OTOC}$, with the stabilizer Renyi entropy, $M_2$, which is well-defined for even-dimensional Hilbert spaces. To evaluate the stabilizer Renyi entropy we use Eq. (1) for $q = 2$ and $n = 2$. The authors of Ref. [19] have shown the relation of the stabilizer Renyi entropy with 8-OTOC. The main difference between our approach with the existing analytical formula in Ref. [19] is the random sampling of a constant number of OTOCs as opposed to the exponential scaling of the number of 8-OTOC terms in the Renyi entropy formula [19].

We use the same random circuits as described in the previous subsection (see Fig. 1), this time for qubits. This way we obtain a comparison between $\delta_{OTOC}$ and the exact stabilizer Renyi entropy. In Fig. 3, we show OTOC fluctuations as a function of mean Renyi entropy, $\overline{M_2}$ and find a dependence corresponding to the fit $\overline{M_2} \approx -1.38\log_2\overline{\delta_{OTOC}} + 0.51$. We repeat the process 10 times to average over different $\delta_{OTOC}$ to obtain statistical error bars. The circuit used for Fig. 3 is a 12 qubit t-doped Clifford and we calculate $\overline{M_2}$ from 10 random instances. Each point in Fig. 3 belongs to a certain number of T-gates in the circuit, $N_T \in [0, 26]$. We note that the range of $N_T$ was motivated by the fact that it has been shown that we need more than or equal to $2N$ T-gates to saturate the magic [42]. We see that regardless of the number of T-gates (and hence the amount of magic in the circuit), our ability to approximate the stabilizer Renyi entropy using OTOC fluctuations remains similar.

For the explanation of the relation observed in Fig. 3, we formulate the following lemma:

**Lemma 1.** Let $M_2(|V\rangle)$ be the stabilizer entropy of the Choi state [43], $|V\rangle \equiv \mathbb{I} \otimes V |I\rangle$ where $|I\rangle \equiv 2^{-N/2} \sum_{i=1}^{2^N} |i\rangle \otimes |i\rangle$, associated to the unitary $V$ and $d = 2^N$, then

$$\mathbb{E}_C \delta_{\text{OTOC}}(U) = \left(\frac{d^2}{d^2-1}\right)^2 2^{-M_2(|V\rangle)} - \frac{2d^2}{(d^2-1)^2} . \tag{7}$$

*Proof.* See Appendix C.

From the Lemma 1 and the numerical results in Fig 3, we can conclude that sampling OTOC fluctuations could lead to more efficiency in measuring $M_2(|V\rangle)$.

For the case of random t-doped Clifford circuits, it generally holds that

$$\mathbb{E}_{C_t} 2^{-M_2(|C_t\rangle)} = \mathbb{E}_{C_t} 2^{-M_2(C_t|0\rangle)} + O(d^{-1}) . \tag{8}$$

Therefore, in the case of a t-doped Clifford circuit, there is no distinction between the stabilizer entropy of $V|0\rangle$ and $|V\rangle$ for sufficiently large $d$.

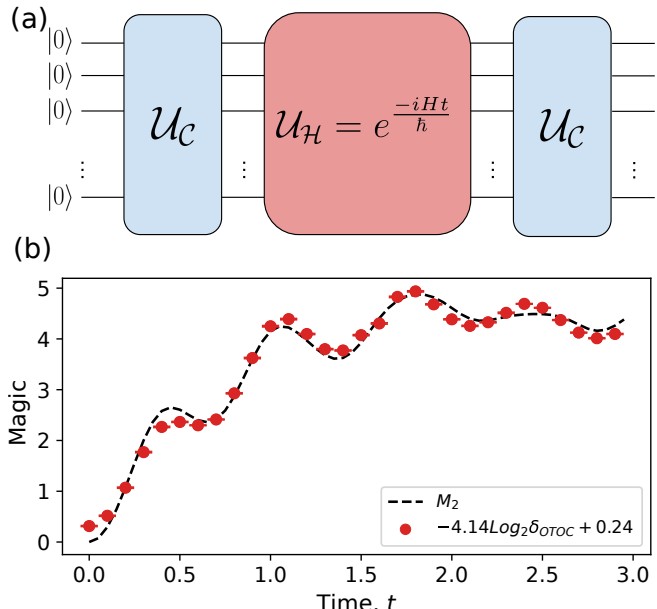

Figure 4: (a) The schematic structure of measuring magic of a time evolution of the Hamiltonian, $\mathcal{U}_{\mathcal{H}}$. The protocol consists of two random Clifford blocks, before and after the desired time evolution block. (b) The comparison of OTOC fluctuations (blue dots) with the exact stabilizer Renyi entropy density (dashed line). The simulation is done for 10 qubits for the Choi state of a chain of length 5.

It is worth noting that the relation of 4-OTOC fluctuations with the averaged 8-OTOC has been studied in Ref. [44]. In contrast, here we describe the relationship to 2-Renyi entropy.

## 3.3 The magic generated by time evolution of a Hamiltonian

In this section, we propose a protocol to measure the magic generated by time evolution under the general Hamiltonian. The time evolution unitary operator of a general time-independent Hamiltonian is a fixed operator. Since, in our method, scrambling is an essential feature, in order to have such a low number of samples we need to create diversity in measured instances of OTOC by introducing additional randomisation in the circuit. We achieve this goal by including two extra blocks of random Clifford circuits, one before the time evolution and one after, see Fig. 4a. Since Clifford gates do not produce any magic by definition, we do not lose any generality for the circuit's magic calculation, but importantly we enhance the scrambling. It is important to keep in mind that the depth of the random Clifford circuit needs to be sufficient to fully scramble the state.

Here, as an example, we consider the Hamiltonian of the transverse-field Ising Hamiltonian,

$$H = -J \sum_i Z_i Z_{i+1} - h \sum_i X_i . \tag{9}$$

The system is in the open boundary condition and $Z_i$ and $X_i$ are the Pauli matrices on the $i$-it qubit. For this simulation, we fix $J = 1$ and $h = 0.5$. The schematic structure of the circuit we consider for its time evolution is shown in Fig. 4(a).

We are considering a chain of $N = 5$ for the Hamiltonian of Eq.9 and the Choi isomorphism, $|V\rangle \equiv \mathbb{I} \otimes V |I\rangle$ where $V = \mathcal{U}_{\mathcal{C}}\mathcal{U}_{\mathcal{H}}\mathcal{U}_{\mathcal{C}}$ and $|I\rangle \equiv 2^{-N/2} \sum_{i=1}^{2^N} |i\rangle \otimes |i\rangle$. From Fig.4(b) we can see that $M_2$ for the local Hamiltonian of the Eq.9 and OTOC fluctuations show similar behaviour,

although the prediction accuracy is lower than for the random t-doped circuits. In this simulation, we used 5 qubits with 50 instances of sampled OTOCs. The time evolves for a total time of $3/J$. We see the same trend of increase in magic in the early time and oscillatory behaviour and stabilization in both stabilizer Renyi entropy and approximated OTOC fluctuations. We used the Qiskit package [45] for this simulation.

## 4 Conclusion and discussion

We have shown aspects of the relation between mana and random sampling of OTOC fluctuations for t-doped circuits which were previously unexplored. In addition to that, we provided numerical evidence that OTOC fluctuation sampling in the scrambled circuits is useful for measuring magic. We were able to mirror behavior for stabilizer Renyi entropy and for mana with significantly lower number measurements. Since the structure of the random circuits is challenging to scale for $N$ qubits, the scalability of this method remains inconclusive, but for up-to 12 qubits we obtained remarkably precise magic estimate with constant number of samples. We also observed that the relation of $\delta_{OTOC}$ and magic is not universal, it showed $\log_2$ behaviour for 2-Renyi entropy and linear behaviour for mana.

Ref. [31] puts forward a statement that the fluctuations of OTOC are always smaller or equal to a specific type of magic monotone. In this work, we complement this statement by numerically showing the relation of OTOC fluctuations to the stabilizer Renyi entropy. We analyzed the accuracy of stabilizer Renyi entropy approximation as a function of the number of samples drawn from scrambled random circuits. While the majority of our simulated data points fulfil the inequality derived in Ref. [31], it is not always the case. This observation is an interesting starting point for further investigation. Also, the analytical relations here could be the starting point for the investigation of the relation between the stabilizer Renyi entropy and the introduced magic measure in Ref. [31].

Additionally, we also extended the method of sampling scrambling random circuits to approximate magic to Hamiltonian evolution and numerically calculated magic for the time evolution governed by an Ising Hamiltonian in a transverse field with very good results in comparison with stabilizer Renyi entropy of the Choi state of the time evolved state of the Hamiltonian. Interestingly the reached agreement is lesser than that of t-doped circuits, but our method still captures general trends of magic behavior during Hamiltonian evolution.

Interesting research direction going forward is to combine our sampling approach with experiment [30] or approximate numerical methods such as tensor networks [32,46] and neural networks [33]. Our method can be used alongside or as a complement to other existing approximation methods [47–52]. Specifically, the algorithm introduced in [48] is an efficient method for measuring Tsallis stabilizer entropy which has a direct relation to stabilizer Renyi entropy. In Ref. [53] lower number of samples comes with doubling the dimension of the Hilbert space. The fact the analytical relationship in Eq. (7) between $\delta_{OTOC}$ and Stabilizer Renyi entropy involves Choi state of a unitary operator might hint at a possible link between these approaches.

All code required to reproduce results presented in this manuscript is available at [54].

## Acknowledgments

We thank Lorenzo Leone for extremely productive peer review process, insightful discussions and the proof of Lemma 1.

## A  Mana

One of the magic monotones is known as *mana*. The restriction of mana is that it is only well-defined for odd prime-dimensional Hilbert spaces. Here, we introduce it for q-dim Hilbert spaces [10] with q an odd prime number. To show how to calculate mana, we first need to define the clock and shift operators corresponding to the q-dimensional Pauli $Z$ gate and Pauli $X$ gate [38],

$$Z = \sum_{n=0}^{q-1} \omega^n |n\rangle \langle n|, \qquad X = \sum_{n=0}^{q-1} |n+1 \, mod \, q\rangle \langle n|, \tag{A.1}$$

with $\omega = e^{2\pi i/q}$. The other necessary definition is the Heisenberg-Weyl operators in prime dimensions,

$$T_{aa'} = \omega^{-2^{-1}aa'} Z^a X^{a'}, \tag{A.2}$$

where $2^{-1} = \frac{q+1}{2}$ (the multiplicative inverse of 2 mod q) and $(a, a') \in \mathbb{Z}_q \times \mathbb{Z}_q$. By following this definition, we can define Pauli strings as

$$T_{\mathbf{a}} = T_{a_1 a_1'} \otimes T_{a_2 a_2'} ... \otimes T_{a_N a_N'}. \tag{A.3}$$

Now, we can define a new basis set for the Hilbert space, known as phase space point operators,

$$A_{\boldsymbol{b}} = q^{-N} T_{\boldsymbol{b}} \left[ \sum_{\boldsymbol{a}} T_{\boldsymbol{a}} \right] T_{\boldsymbol{b}}^{\dagger}, \tag{A.4}$$

and these phase space point operators form a complete basis set for $\mathbb{C}^{q^N \otimes q^N}$. Thus, we can expand any density matrix $\rho$ in this basis,

$$\rho = \sum_{\mathbf{u}} W_\rho(\mathbf{u}) A_{\boldsymbol{u}}. \tag{A.5}$$

The coefficients $W_\rho(\mathbf{u})$ are called discrete Wigner functions and we can define mana as

$$\mathcal{M}(\rho) = \log \sum_{\mathbf{u}} \left| W_\rho(\mathbf{u}) \right|. \tag{A.6}$$

As we already stated in the main text, we are dealing with Clifford and non-Clifford operations. The Clifford gates map Pauli strings to other Pauli strings, up to an arbitrary phase [55],

$$C = \left\{ U : U T_{\boldsymbol{a}} U^{\dagger} = e^{i\phi} T_{\boldsymbol{b}} \right\}. \tag{A.7}$$

Since the Clifford gates map each of these Pauli strings to each other, each Clifford unitaries also map the computational basis to one of the eigenstates of Pauli strings. These eigenstates are called stabilizer states. Since stabilizer states are prepared with only Clifford gates, their mana is *zero*.

## B  Clifford and non-Clifford gates definitions

In this appendix, we are introducing the gates that we have used in this study. We introduce both 2-dimensional Hilbert spaces and higher-dimensional Hilbert spaces.

### B.1 Clifford gates

The set of Clifford gates is the second level of Clifford hierarchy [9] that are the following gates in 2-dimensional Hilbert spaces,

$$H_2 = \frac{1}{\sqrt{2}} \begin{bmatrix} 1 & 1 \\ 1 & -1 \end{bmatrix}, \quad P_2 = \begin{bmatrix} 1 & 0 \\ 0 & i \end{bmatrix},$$

$$\text{CNOT} = |0\rangle \langle 0| \otimes I + |1\rangle \langle 1| \otimes X. \tag{B.1}$$

The generalization of these gates is straightforward [56]. The d-dimensional Hadamard gate, $H_d$, is

$$H_d |j\rangle = \frac{1}{\sqrt{d}} \sum_{i=0}^{d-1} \omega^{ij} |i\rangle, \quad j \in \{0, 1, 2, ..., d-1\}, \tag{B.2}$$

where $\omega := e^{2\pi i/d}$. The next gate is the d-dimensional Phase gate, $P_d$,

$$P_d |j\rangle = \omega^{j(j-1)/2} |j\rangle, \tag{B.3}$$

and, finally, the generalized CNOT gate that is known as $CSUM_d$ gate and defined as

$$CSUM_d |i, j\rangle = |i, i + j(\bmod d)\rangle, \quad i, j \in \{0, 1, 2, ..., d-1\}. \tag{B.4}$$

### B.2 Non-Clifford gates

Clifford gates are not sufficient for universal quantum computation and we at least need one non-Clifford gate to have this universality [57,58]. One of these gates is the T-gate that emerges from the third level Clifford hierarchy. The definition of T-gate for 2-dimensional Hilbert space is

$$T_2 = \begin{bmatrix} 1 & 0 \\ 0 & e^{i\pi/4} \end{bmatrix}. \tag{B.5}$$

The generalization of T-gate to higher dimensional Hilbert spaces is not so straightforward [38]. Here, we only write down the matrices of the T-gate for 3-dimensional Hilbert spaces which are useful for us. The 3-dimensional Hilbert space T-gate is

$$T_3 = \begin{bmatrix} 1 & 0 & 0 \\ 0 & e^{2\pi i/9} & 0 \\ 0 & 0 & e^{-2\pi i/9} \end{bmatrix}. \tag{B.6}$$

## C Proof of lemma 1

In order to show lemma 1, we need to have a close look at the first term in $\delta_{\text{OTOC}}$,

$$\mathbb{E}_C \text{OTOC}^2(U) = \int dC_1 dC_2 \frac{1}{d^2} \operatorname{tr}\left(T_{(12)(34)} V^{\dagger \otimes 4} C_1^{\dagger \otimes 4} A^{\otimes 4} C_1^{\otimes 4} V^{\otimes 4} C_2^{\otimes 4} B^{\otimes 4} C_2^{\dagger \otimes 4}\right). \tag{C.1}$$

By averaging over $C_1$ we will have

$$\int dC_1 C_1^{\dagger \otimes 4} A^{\otimes 4} C_1^{\otimes 4} = \frac{1}{d^2 - 1} \sum_{P \in \mathbb{P}_n \setminus \{\mathbb{I}\}} P^{\otimes 4}. \tag{C.2}$$

By averaging over Clifford circuits we will get a flat distribution over the Pauli group $\mathbb{P}_n$ but the identity. By defining $Q := d^{-2} \sum_{P \in \mathbb{P}_n} P^{\otimes 4}$, the Eq. C.2 becomes

$$\int dC_1 C_1^{\dagger \otimes 4} A^{\otimes 4} C_1^{\otimes 4} = \frac{d^2}{d^2 - 1} Q - \frac{1}{d^2 - 1} \mathbb{I}^{\otimes 4}, \tag{C.3}$$

we get similar results for averaging over $C_2$ on the non-identity Pauli operator $B$. So Eq. C.1 would become

$$\mathbb{E}_C \text{OTOC}^2(U) = \frac{1}{d^2} \left( \frac{d^2}{d^2 - 1} \right)^2 tr \left( Q V^{\otimes 4} Q V^{\dagger \otimes 4} \right) - \frac{2d^2 - 1}{(d^2 - 1)^2}, \tag{C.4}$$

where we used the fact that $\text{tr}(\mathcal{O} Q T_{(12)(34)}) = \text{tr}(\mathcal{O} Q)$ for every $\mathcal{O}$ [42]. From Ref. [25] we know that the average $\mathbb{E}_C \text{OTOC}(U) = -(d^2 - 1)^{-1}$. We know that from Ref. [59] the second stabilizer Renyi entropy of the Choi state of the unitary $V$ is

$$M_2(|V\rangle) = -\log \frac{1}{d^2} \text{tr}\left( Q V^{\otimes 4} Q V^{\dagger \otimes 4} \right). \tag{C.5}$$

So the final equation would be

$$\mathbb{E}_C \delta_{\text{OTOC}}(U) = \left( \frac{d^2}{d^2 - 1} \right)^2 2^{-M_2(|V\rangle)} - \frac{2d^2}{(d^2 - 1)^2}. \tag{C.6}$$

We see that from Eq. 7 the second stabilizer Renyi entropy is related to the Choi state $|V\rangle$ associated with the unitary $V$ and not the second stabilizer Renyi entropy of the state $V|0\rangle$. Fortunately, in the case of $V$ being a random t-doped circuit, $C_t$, from Ref. [59] we have

$$\mathbb{E}_{C_t} \delta_{\text{OTOC}}(C_t)$$
$$= \frac{d^4}{(d^2 - 1)^2} \left[ \frac{4(6 - 9d^2 + d^4)}{d^4(d^2 - 9)} + \frac{d^2 - 1}{d^2} \left( \frac{(d + 2)(d + 4)f_+^t}{6d(d + 3)} + \frac{(d - 2)(d - 4)f_-^t}{6d(d - 3)} + 2\frac{(d^2 - 4)\left( \frac{f_+ + f_-}{2} \right)^t}{3d^2} \right) \right]$$
$$- 2\frac{d^2}{(d^2 - 1)^2}, \tag{C.7}$$

where

$$f_\pm = \frac{3d^2 \mp 3d - 4}{5(d^2 - 1)}, \tag{C.8}$$

for d being large we have

$$\mathbb{E}_{C_t} \delta_{\text{OTOC}}(C_t) = \left( \frac{3}{4} \right)^t + O(d^{-2}). \tag{C.9}$$

In Ref. [19], the average value of 2-stabilizer entropy over a t-doped Clifford circuit is given as

$$-\log \left( \frac{4 + (d - 1)f_+^t}{3 + d} \right) \le \mathbb{E}_{C_t} M_2(C_t |0\rangle) \le \begin{cases} t, & t < N - 1, \\ N - 1. \end{cases} \tag{C.10}$$

From Eq. C.7 and Eq. C.10, it is straightforward to show that for a random t-doped Clifford circuit,

$$\mathbb{E}_{C_t} 2^{-M_2(|C_t\rangle)} = \mathbb{E}_{C_t} 2^{-M_2(C_t |0\rangle)} + O(d^{-1}). \tag{C.11}$$

# D  Propagation of error

Let us analyze how errors propagate in Eq.7. We can write the error in $M_2$ in terms of $\delta_{OTOC}$ as

$$\Delta M_2 = \frac{\partial M_2}{\partial \delta_{OTOC}} \Delta(\delta_{OTOC}). \tag{D.1}$$

At the same time, we can rewrite Eq.7 as

$$M_2 = -\log_2 \frac{\mathbb{E}\delta_{OTOC} + \beta}{\alpha}. \tag{D.2}$$

This formula allows us to evaluate the derivative on the right-hand side of (D.1) as

$$\frac{\partial M_2}{\partial \delta_{OTOC}} = -\frac{1}{(\mathbb{E}\delta_{OTOC} + \beta)\ln 2}. \tag{D.3}$$

Combining (D.1) and (D.3) we obtain

$$\Delta M_2 = -\frac{1}{(\mathbb{E}\delta_{OTOC} + \beta)\ln 2} \Delta(\delta_{OTOC}). \tag{D.4}$$

Error in $M_2$ is thus proportional to the error in $\delta_{OTOC}$ with an inverse factor of the expectation value of $\delta_{OTOC}$.

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
