# Peer review of "Quantifying non-stabilizerness via information scrambling"

_SciPost Physics, doi:SciPost Phys. 16, 043 (2024)_

## Round 3 · Referee Report · Anonymous (Referee 1) · 2023-8-21

Report

This work proposes to measure nonstabilizerness using out-of-time-order correlators.
The authors provide numerical evidence that for two classes of dynamics (Clifford + T and ising) that nonstabilizerness and OTOC fluctations behave similarly. From this numerical fitting, it is proposed to use OTOC as a proxy to measure nonstabilizerness in experiment.

This method rests on their numerical studies to show OTOC fluctuation and stabilizer entropy (for n=2) behave similarly. However, this may not be sufficient evidence. First, it is known that stabilizer entropy behaves differently depending on the chosen index n (e.g. arXiv:2303.10152). Further, the results rest on two numerical studies of two specific models, however it is not clear whether stabilizer entropy and OTOC behave similar in general, and it is possible that for other models the equivalence may not hold.
Further, the claimed advantage of OTOC over stabilizer entropy in terms of scalability may not hold given recent results (e.g. [47]).

This works provides an interesting link between OTOCs and nonstabilizerness, and makes progress on the problem of measuring nonstabilizerness. It is written in a clear way, with a sufficient introduction and details. Citations to most literature are given, though there are mistakes in the formatting.
The code is not available, but sufficient information to reproduce the results are given. The conclusion summarizes the results, though evidence for two major points (OTOC scale better than measuring stabilizer entropy directly, as well as OTOC behave similar to nonstabilizerness) are not sufficiently evidenced.

Requested changes

  • The key statement is the similarity between OTOC and stabilizer entropy for n=2. This is a key point, but numerical results on two models does not seem sufficient. To strengthen this claim, I suggest to search for possible counter-examples regarding whether OTOC and stabilizer entropy do behave differently. This may help to strengthen the claims.
  • Complexity of OTOC vs stabilizer entropy: I suggest to discuss the scaling of measuring OTOC and stabilizer entropy more in depth given recent results for both experiment ([47]) and numerics (arXiv:2209.10541), and do a proper comparison of complexity and experimental protocols. This discussion should give evidence whether OTOC is indeed better in some instances for measuring nonstabilizerness.

  • "Interestingly, in the context of quantum error correction, the Clifford gates can be implemented fault-tolerantly": Note that this is not true for all error correction codes, some codes can implement non-Clifford gates transversally, but not all Cliffords.

  • page 3: "Another magic monotone introduced for qubits is the Stabilizer Renyi Entropy": note that stabilizer netropy not a monotone (i.e. it is not defined for mixed states, and also not a monotone for pure states as channels that map pure to pure states for n<2 it has been shown not to be a monotone (arXiv:2303.10152))
  • page 5: Why does the ising model evolution is surrounded by Clifford gates? Why is this necessary? This should be explained more clearly.
  • it is noted that stabilizer entropy can only be computed up to 12 qubits, however computations up to 15 qubits have been performed (arXiv:2209.10541). Thus, I am not sure the claim that computing OTOC is easier than stabilizer entropy is valid.
  • It is mentioned in the conclusion that OTOCs are better to compute nonstabilizerness due to scalability. OTOCs can be measured efficiently by sampling, however similarly it has been noted that stabilizer entropy can be measured efficiently as well by sampling (e.g. [47]). In what sense is thus OTOC better than stabilizer entropy?
  • references are not properly formatted and missing important info, e.g. 2, 8, 31, 47, 48
  • I am wondering whether the equivalence of OTOC and nonstabilizerness could be made analytical, e.g. exact equivalence was found for the inverse participation ratio recently (e.g. see arXiv:2305.11797)

  • validity: good
  • significance: ok
  • originality: good
  • clarity: high
  • formatting: good
  • grammar: excellent

Author:  Arash Ahmadi  on 2023-11-15  [id 4115]

(in reply to Report 1 on 2023-08-21)

We thank the referee for their assessment. We will reply to all the questions in detail below. Here we want to directly address the code availability to avoid any misunderstandings: the code is available and mentioned in the manuscript as Ref. 54.

Requested changes

  • The key statement is the similarity between OTOC and stabilizer entropy for n=2. This is a key point, but numerical results on two models does not seem sufficient. To strengthen this claim, I suggest to search for possible counter-examples regarding whether OTOC and stabilizer entropy do behave differently. This may help to strengthen the claims. We fully agree with the point of the reviewer here. Thanks to reviewer number three, Dr. Lorenzo Leone, now there is an analytical relation to 2-stabilizer entropy. As you mentioned here the relation of OTOC is not generally true and could differ between different methods, as we have shown here for mana it is 1-δotoc and for M2 it is log2 δotoc. We mentioned this point in section IV, paragraph one.

  • Complexity of OTOC vs stabilizer entropy: I suggest to discuss the scaling of measuring OTOC and stabilizer entropy more in depth given recent results for both experiment ([47]) and numerics (arXiv:2209.10541), and do a proper comparison of complexity and experimental protocols. This discussion should give evidence whether OTOC is indeed better in some instances for measuring nonstabilizerness. We appreciate the reviewer's point on adding discussion on the recent papers. We included a comprehensive discussion about the similarities and the potential relationship between OTOC fluctuations and Tsallis stabilizer entropies on section IV paragraph 4.

  • "Interestingly, in the context of quantum error correction, the Clifford gates can be implemented fault-tolerantly": Note that this is not true for all error correction codes, some codes can implement non-Clifford gates transversally, but not all Cliffords. We thank the reviewer for pointing out this important point. We adjusted the language accordingly.

  • page 3: "Another magic monotone introduced for qubits is the Stabilizer Renyi Entropy": note that stabilizer entropy not a monotone (i.e. it is not defined for mixed states, and also not a monotone for pure states as channels that map pure to pure states for n<2 it has been shown not to be a monotone (arXiv:2303.10152)) We thank the reviewer for pointing out this important point. We changed the phrasing.

  • page 5: Why does the Ising model evolution is surrounded by Clifford gates? Why is this necessary? This should be explained more clearly. We thank the reviewer for raising this point. For our method to work, scrambling is an essential feature. Clifford blocks fulfil this role in the case of a random circuit. To the best of our understanding, the method we present in the manuscript requires such a low number of samples because we measure the circuit in the regime where the initial state is fully scrambled. Hamiltonian evolution alone does not necessarily have this feature and without the scrambling we do not have enough diversity in measured samples to be able to sample only a small subset of instances. We added more detailed explanation in paragraph one of section III.C.

  • it is noted that stabilizer entropy can only be computed up to 12 qubits, however computations up to 15 qubits have been performed (arXiv:2209.10541). Thus, I am not sure the claim that computing OTOC is easier than stabilizer entropy is valid. We thank the reviewer for mentioning this reference and we apologise for overlooking it in our literature search. We adjusted our claims accordingly and added the reference.

  • It is mentioned in the conclusion that OTOCs are better to compute nonstabilizerness due to scalability. OTOCs can be measured efficiently by sampling, however similarly it has been noted that stabilizer entropy can be measured efficiently as well by sampling (e.g. [47]). In what sense is thus OTOC better than stabilizer entropy? We agree with the referee’s point. In the present manuscript we can only show analytically the logarithmic relation to 2-Renyi entropy. At the same time, we provide strong numerical evidence that, at least for a classically simulatable number of qubit, the number of OTOC samples we need to exactly approximate Stabilizer Renyi Entropy and Mana is extremely low. For instance, for 12 qubits, where we would have to average 4^12 OTOC to satisfy the analytical formula, we obtain correct scaling and negligible numerical error with as little as 50 OTOC samples (see Figure 2 and 3). We admittedly do not provide analytical proof for this behaviour, but after thorough numerical testing, we wish to highlight the property that in practice, when sampling from a scrambled state, significantly lower number of samples is needed compared to what analytical relationship would suggest. We reformulated our conclusions accordingly.

  • references are not properly formatted and missing important info, e.g. 2, 8, 31, 47, 48 We thank the referee for pointing out the formatting issue. We corrected this.

  • I am wondering whether the equivalence of OTOC and nonstabilizerness could be made analytical, e.g. exact equivalence was found for the inverse participation ratio recently (e.g. see arXiv:2305.11797) We again acknowledge contribution of Dr. Lorenzo Leone during this peer-review process. We added the analytical relationship into the manuscript and provided detailed proof in Appendix C.

---

## Round 3 · Referee Report · Anonymous (Referee 2) · 2023-8-28

Strengths

-An interesting connection between magic monotones and quantum information scrambling is found. This link may have a practical importance in future.
-A gitlab page containing the code used for the numerical simulations is provided

Weaknesses

-The numerics should be improved, for instance addressing the scaling of required samples with the system size $N$.
-Some of the statement and messages are not very clear.

Report

The paper should be accepted by SciPost after some minor revisions and a general improvement of the clarity of the work (see the paragraph Requested changes).

Requested changes

Questions/remarks/observations:
-Regarding your sentence "[Mana is] defined for odd-dimensional local Hilbert space and thus not possible to evaluate for qubits": from a technical/mathematical point of view, what does make mana not well-defined for an even dimensional Hilbert spaces? For clarity, I think you should comment on this in the text.
-Regarding Figure 2, how did you compute exactly the vertical error bars? In the caption, you write ``The vertical error bar is the statistical error calculated as the inverse of the square root of the number of samples". However, I guess that the statistical error associated with the standard deviation of OTOC should be related to the fourth momentum. Am I wrong? Regarding the horizontal error bars: did you include here the factor $1/\sqrt{N_{samples}}$ coming from the fact you are estimating the fluctuations of the mean? Similar questions for the error bars in Figure 3.
-Do you think your analysis would be affected by considering other operators for the OTOC? Maybe you can add a comment about this. Also, for which reason you considered operators localized on the boundaries ($X_{N-1}$ and $Z_1$)? [By the way, if the chain has length $N$, I think the last site should be labelled by $N$...]
-I do not understand well the purpose of Figure 3$c)$: are you claiming that you can estimate the second Stabilizer Renyi Entropy (SRE) with reasonable accuracy by increasing the number of sampled OTOC? In this case, I think you should perform a better analysis with increasing values of the number of samples.Also, why are you now forgetting about your linear fit coefficients and just estimating the average SRE as $1 - \delta_{OTOC}$? And why you consider the observation that ``the absolute error for both sample counts is higher in the intermediate region of the T-gate count" as notable?
-I think your statement that "OTOC flcutuations is the numerically lesser demanding method to estimate magic" should be made more precise. In particular, you have to address the scaling of the number of samples needed to reach a given accuracy with the system size $N$.
-What is the main message of Figure 5 and in general of Section C? Indeed, you are not finding a perfect agreement between OTOC fluctuations of the circuit in Figure 5$a)$ and the SRE of the time evolved state, just a similar behaviour... and so? Are you claiming that OTOC fluctuations are magic monotone as well?

Other proposed changes:
-Am I wrong or the letter $q$ to indicate the local Hilbert space dimension is defined \textit{after} using it? I think it should be better to introduce the notation before of using it.
-I think that when you report the results of a (linear) fit you should not use the symbol $=$, because it is not an exact mathematical equality. You can use $\simeq$ instead.
-I think a reader would benefit from having the following additional references: https://arxiv.org/abs/2303.05536 and https://arxiv.org/abs/2304.01175. The first paper introduces a new method to evaluate efficiently the SRE for a Matrix Product State, by performing a Pauli sampling. I think it should be cited around the sentence "approximate numerical methods such as tensor networks". The second one regards a way of connecting the SRE with the properties of the entanglement spectrum, and I think can be considered as a general reference about SRE.

  • validity: good
  • significance: good
  • originality: good
  • clarity: ok
  • formatting: good
  • grammar: good

Author:  Arash Ahmadi  on 2023-11-15  [id 4116]

(in reply to Report 2 on 2023-08-28)

Requested changes
Questions/remarks/observations:
-Regarding your sentence "[Mana is] defined for odd-dimensional local Hilbert space and thus not possible to evaluate for qubits": from a technical/mathematical point of view, what does make mana not well-defined for an even dimensional Hilbert spaces? For clarity, I think you should comment on this in the text.

We thank the reviewer for highlighting this. We commented on why mana is not calculable for qubits in section III. B paragraph one.

-Regarding Figure 2, how did you compute exactly the vertical error bars? In the caption, you write The vertical error bar is the statistical error calculated as the inverse of the square root of the number of samples". However, I guess that the statistical error associated with the standard deviation of OTOC should be related to the fourth momentum. Am I wrong? Regarding the horizontal error bars: did you include here the factor 1/√Nsamples coming from the fact you are estimating the fluctuations of the mean? Similar questions for the error bars in Figure 3.

We thank the referee for pointing this out. Previously, our error bar was precisely the statistical error of 1/sqrt(N), where N is total number of samples. This statistical behaviour dominates the error. Prompted by referee’s question we adjusted the error bar by repeating δotoc estimation 10 times and considered the the std/sqrt( 10) of these different instances, thus including both standard deviation from the mean and statistical error. As seen in Figures 2 and 3, this new estimation decreased overall error bar significantly. We thank the referee for the suggestion.

-Do you think your analysis would be affected by considering other operators for the OTOC? Maybe you can add a comment about this. Also, for which reason you considered operators localized on the boundaries (XN−1 and Z1)? [By the way, if the chain has length N, I think the last site should be labelled by N...]

As long as the two operators, in our paper A and B, are commuting, these results hold, meaning we could apply these (or other Paulis) on any qubit pair in our circuit. We clarified our setup in Section II. B paragraph one.

The question about different types of correlators is very interesting. Here, we considered OTOC due to their interesting analytical relationship with the magic monotones we are attempting to approximate (proof added in revision, section IIIB). At the same time, we see that numerically, we can significantly undersample the analytical relationship of these two quantities and still obtain a very precise approximation.
Designing other correlators that would have this property goes beyond the scope of what we did here, but it is nonetheless very stimulating question that might be addressed in the future research.

-I do not understand well the purpose of Figure 3c): are you claiming that you can estimate the second Stabilizer Renyi Entropy (SRE) with reasonable accuracy by increasing the number of sampled OTOC? In this case, I think you should perform a better analysis with increasing values of the number of samples.Also, why are you now forgetting about your linear fit coefficients and just estimating the average SRE as 1− δOTOC? And why you consider the observation that the absolute error for both sample counts is higher in the intermediate region of the T-gate count" as notable?
We fully agree with the referee. In the new version of the manuscript we omitted this figure and added the analytical results instead.

-I think your statement that "OTOC fluctuations is the numerically lesser demanding method to estimate magic" should be made more precise. In particular, you have to address the scaling of the number of samples needed to reach a given accuracy with the system size N.
The referee is completely correct at pointing out this challenge and it in fact highlights the central challenge in the field of non-stablizerness. Calculating non-stabilizerness measures is extremely numerically costly both for exact and for approximated quantum states, see Refs. [19, 55]. Our algorithm, with it’s beneficial scaling in number of OTOC instances is attempting to solve this challenge, but at the moment we are not yet able to show large system scaling beyond 12 qubits.
Interesting question attached to this challenge is that the approximative methods do not necessarily maintain the entanglement or non-stabilizerness structure of the state and this is a key challenge with deploying them. We are working actively towards scaling that is at the moment unfortunately beyond the scope of this manuscript.

-What is the main message of Figure 5 and in general of Section C? Indeed, you are not finding a perfect agreement between OTOC fluctuations of the circuit in Figure 5a) and the SRE of the time evolved state, just a similar behaviour... and so? Are you claiming that OTOC fluctuations are magic monotone as well?

We thank the referee for highlighting this point. In the first part of the manuscript we analyze the behavior of our method for discretized t-doped circuit setting. We were interested to see if it’s possible to formulate corresponding statements about OTOC sampling as a magic estimation for continuous Hamiltonian evolution as well. Indeed we find that with just a handful OTOC samples we can estimate magic generated by continuous time-evolution if less precisely. Note that we added the stabilizer blocks to achieve the scrambling that allows us to sample less. So indeed, we are claiming that averaging over low number of OTOC samples can play a role of magic monotone even in the continuous time evolution setting.

We reformulated paragraph 1 of section III.C to reflect this explanation. We thank the referee for pointing out the lack of clarity.

Other proposed changes:
-Am I wrong or the letter q to indicate the local Hilbert space dimension is defined \textit{after} using it? I think it should be better to introduce the notation before of using it.
We thank the referee for catching this mistake, we explained the notation in the first paragraph of section III.A.

-I think that when you report the results of a (linear) fit you should not use the symbol =, because it is not an exact mathematical equality. You can use ≃ instead.
We thank the referee, we changed the notation.

-I think a reader would benefit from having the following additional references: https://arxiv.org/abs/2303.05536 and https://arxiv.org/abs/2304.01175. The first paper introduces a new method to evaluate efficiently the SRE for a Matrix Product State, by performing a Pauli sampling. I think it should be cited around the sentence "approximate numerical methods such as tensor networks". The second one regards a way of connecting the SRE with the properties of the entanglement spectrum, and I think can be considered as a general reference about SR

We thank to the referee for pointing out missing literature. We reviewed these references and added them to the manuscript.

Anonymous on 2023-12-12  [id 4184]

(in reply to Arash Ahmadi on 2023-11-15 [id 4116])

The paper quality improved a lot after revision. In particular, Lemma 1 is now a very interesting and clear result. However, I still do not understand well the statements of Sec.III C, particularly in connection with Fig.4. What is the cause of the discrepancy between M2 and -Log_2[OTOC fluctuations]? There shouldn't be an exact correspondence (a part for statistical errors) due to Eq.7? I don't understand the utility of the procedure applied to continuous time evolution, given that it does not reproduce accurately M2. Apart from these aspects, I believe that the work has reached a sufficient level of clarity to deserve publication on SciPost.

Anonymous on 2024-01-12  [id 4240]

(in reply to Anonymous Comment on 2023-12-12 [id 4184])

We thank the reviewer who turned our attention to more about the point. We understood we had a small bug in the code that made this gap, it is fixed now and the agreement between the exact magic of the Choi state and OTOC fluctuation is much better.

---

## Round 3 · Referee Report · Lorenzo Leone (Referee 3) · 2023-9-7

Strengths

1) The paper addresses the highly challenging task of establishing a connection between magic and information scrambling. While there had been hints suggesting this connection, a systematic study like the one presented in this paper was absent.

2) The author's decision to investigate scrambling fluctuations is indeed well-suited for this purpose. Magic is not related to scrambling per se, being Clifford unitaries vey good scramblers of information. However, it extends beyond mere scrambling and delves into the intricacies of scrambling complexity, effectively captured by ensemble fluctuations (with respect to the Clifford average, as pointed out in the author's study). This complexity of scrambling plays a crucial role in determining whether the ensemble of unitaries possesses universal properties or not, and ultimately, it manifests in finer-grained aspects of scrambling, such as those exemplified by the 8-point OTOCs.

Weaknesses

The paper's weakness lies in the lack of detailed numerical analysis, which remains incomplete. Given that the authors assert that measuring 4-OTOC fluctuations can offer a more efficient means of measuring magic compared to previous works, the scaling with N is of paramount significance to give sense this claim and, most importantly, sentences like

'... for the system sizes studied in this work, we have not encountered any evidence
that the sampling complexity should be exponential in
the number of qubits...'

to support the claim of the paper are better to be avoided. However, we are aware that stabilizer entropy can be computed numerically (for entangled states) up to N=12 and thus a finite size scaling becomes challenging.

Report

I recommend the paper for publication, but it is crucial that the authors undertake a significant revision. Although the topic holds great importance, and the chosen figure of merit is appropriate, there is substantial work that needs to be completed before the paper can be deemed ready for publication.

Requested changes

(A) As demonstrated in the attached PDF, the proposed proportionality $\delta_{otoc}=AM_2+B$ between the 4-OTOC fluctuations $\delta_{otoc}$ (with respect to the Clifford average) and the second stabilizer entropy $M_2$ is incorrect. Instead, despite OTOC fluctuations exhibit a positive correlation with the $2$-stabilizer entropy, this relationship follows an exponential dependence upon the 2-stabilizer entropy. While the attached PDF provides analytical proof for this correction, there were already some indicators within the authors' work that hinted at this being the case:

1) Fig. $(3) c$ illustrates that OTOC fluctuations and stabilizer entropy exhibit the most significant differences in the middle range, precisely where a linear line and a saturating exponential curve differ the most, as depicted in Fig. $(1)$ of the attached PDF.

2) I believe that the log-plot of $\delta_{otoc}$ (and not $1-\delta_{otoc}$) in Fig. $(3) b$ would result in the blue and red points aligning along a linear trend.

(B) The paper requires further numerical or analytical investigation to assess the efficiency of measuring magic through OTOC fluctuations. Otherwise, even if the authors are unable to present a finite size scaling that supports their assertion, the link between magic and OTOCs remains intriguing. However, in the latter case, all assertions regarding the efficiency of measuring magic through scrambling would be better to be omitted, including a potential alteration of the paper's title. It is essential to ensure that we do not inadvertently misguide the scientific community. Furthermore, in light of point (A), one can still measure the 2-stabilizer entropy via the OTOC fluctuations, although the functional dependence differs from the one suggested in the paper. It would be better to consider this in a future revision.

(C) I believe that a discussion on Eq.~$(19)$ of (PhysRevA.106.062434) should be added as it is linking the complexity of scrambling, the 8-OTOC and the 4-OTOC fluctuations explored by the authors.

Attachment

  • validity: -
  • significance: -
  • originality: -
  • clarity: -
  • formatting: -
  • grammar: -

Author:  Arash Ahmadi  on 2023-11-15  [id 4117]

(in reply to Report 3 by Lorenzo Leone on 2023-09-07)

Requested changes

(A) As demonstrated in the attached PDF, the proposed proportionality δotoc=AM2+B between the 4-OTOC fluctuations δotoc (with respect to the Clifford average) and the second stabilizer entropy M2 is incorrect. Instead, despite OTOC fluctuations exhibit a positive correlation with the 2-stabilizer entropy, this relationship follows an exponential dependence upon the 2-stabilizer entropy. While the attached PDF provides analytical proof for this correction, there were already some indicators within the authors' work that hinted at this being the case:

We thank the referee for sharing this proof with us and significantly improving our manuscript. We re-fitted the proven analytical relation to our numerical results and indeed it offers much better agreement than our earlier fit.

1) Fig.(3)c illustrates that OTOC fluctuations and stabilizer entropy exhibit the most significant differences in the middle range, precisely where a linear line and a saturating exponential curve differ the most, as depicted in Fig. (1) of the attached PDF. Indeed in the light of the analytical proof we omitted Fig. 3C.

2) I believe that the log-plot of δotoc (and not 1−δotoc) in Fig. (3)b would result in the blue and red points aligning along a linear trend. This is indeed correct and implemented in the new version of the manuscript.

(B) The paper requires further numerical or analytical investigation to assess the efficiency of measuring magic through OTOC fluctuations. Otherwise, even if the authors are unable to present a finite size scaling that supports their assertion, the link between magic and OTOCs remains intriguing. However, in the latter case, all assertions regarding the efficiency of measuring magic through scrambling would be better to be omitted, including a potential alteration of the paper's title. It is essential to ensure that we do not inadvertently misguide the scientific community. Furthermore, in light of point (A), one can still measure the 2-stabilizer entropy via the OTOC fluctuations, although the functional dependence differs from the one suggested in the paper. It would be better to consider this in a future revision.

We thank the reviewer for directing us to this issue. Due to the limits to the exact simulatability of large system sizes, we agree that we cannot support the idea of efficiency by exact simulation. Our present algorithm does, however, show beneficial scaling in the number of OTOC samples compared to the analytical formulation. We highlighted this possible sampling advantage in a revised version without making statements of efficiency.

(C) I believe that a discussion on Eq.~(19) of (PhysRevA.106.062434) should be added as it is linking the complexity of scrambling, the 8-OTOC and the 4-OTOC fluctuations explored by the authors.

We added the discussion of this reference in the manuscript, specifically in the last paragraph of section III. B.

---

## Round 4 · Referee Report · Anonymous (Referee 1) · 2023-12-20

Report

The authors answered my comments and improved the manuscript. The manuscript can be published after adressing following minor comments:

  • In appendix C, the derivation of the relationship between magic of Choi state and otoc fluctuation is not clear. It may be good to expand the proof. In particular, how in equation C5 does the Choi state appear?
  • page 4: Choi state is introduced, but explained only later on page 4. It would be better to define Choi state first
  • page 4: There is an exact relationship between magic and OTOC, yet why is there such a large gap between the curves for OTOC and magic?
  • validity: -
  • significance: -
  • originality: -
  • clarity: -
  • formatting: -
  • grammar: -

Author:  Arash Ahmadi  on 2024-01-12  [id 4239]

(in reply to Report 1 on 2023-12-20)

We defined the relation of the Choi state that appeared in Eq. C5 in equation C6 with the citation to the paper that proved it, however, we acknowledge the reviewer’s point and changed the order of these equations to be clear

We added the definition of the Choi state in an earlier stage on Lemma 1.

We thank the reviewer who turned our attention to more about the point. We understood we had a small bug in the code that made this gap, it is fixed now and the agreement between the exact magic of the Choi state and OTOC fluctuation is much better.

---

## Round 4 · Referee Report · Lorenzo Leone (Referee 3) · 2024-1-2

Report

The authors have significantly improved their manuscript by accurately linking stabilizer entropy and OTOCs, which is insightful. However, I remain concerned about the claim about the scaling of resources. While, in the introduction, the authors mention that it seems more resource-efficient to measure OTOC fluctuations rather than stabilizer entropy based on numerical observations, a clearer theoretical analysis based on Lemma 1 could better demonstrate this advantage. However, I understand that such an analysis might be tricky and possibly beyond the scope of this work.

Overall, I believe that now the manuscript is ready for publication in SciPost.

---

## Round 4 · Author Response

Referee reports:

Anonymous Report 1

Report This work proposes to measure nonstabilizerness using out-of-time-order correlators. The authors provide numerical evidence that for two classes of dynamics (Clifford + T and ising) that nonstabilizerness and OTOC fluctations behave similarly. From this numerical fitting, it is proposed to use OTOC as a proxy to measure nonstabilizerness in experiment. This method rests on their numerical studies to show OTOC fluctuation and stabilizer entropy (for n=2) behave similarly. However, this may not be sufficient evidence. First, it is known that stabilizer entropy behaves differently depending on the chosen index n (e.g. arXiv:2303.10152). Further, the results rest on two numerical studies of two specific models, however it is not clear whether stabilizer entropy and OTOC behave similar in general, and it is possible that for other models the equivalence may not hold. Further, the claimed advantage of OTOC over stabilizer entropy in terms of scalability may not hold given recent results (e.g. [47]). This works provides an interesting link between OTOCs and nonstabilizerness, and makes progress on the problem of measuring nonstabilizerness. It is written in a clear way, with a sufficient introduction and details. Citations to most literature are given, though there are mistakes in the formatting. The code is not available, but sufficient information to reproduce the results are given. The conclusion summarizes the results, though evidence for two major points (OTOC scale better than measuring stabilizer entropy directly, as well as OTOC behave similar to nonstabilizerness) are not sufficiently evidenced. We thank the referee for their assessment. We will reply to all the questions in detail below. Here we want to directly address the code availability to avoid any misunderstandings: the code is available and mentioned in the manuscript as Ref. 54.

Requested changes

  • The key statement is the similarity between OTOC and stabilizer entropy for n=2. This is a key point, but numerical results on two models does not seem sufficient. To strengthen this claim, I suggest to search for possible counter-examples regarding whether OTOC and stabilizer entropy do behave differently. This may help to strengthen the claims.

We fully agree with the point of the reviewer here. Thanks to reviewer number three, Dr. Lorenzo Leone, now there is an analytical relation to 2-stabilizer entropy. As you mentioned here the relation of OTOC is not generally true and could differ between different methods, as we have shown here for mana it is 1-δotoc and for M2 it is log2 δotoc. We mentioned this point in section IV, paragraph one.

  • Complexity of OTOC vs stabilizer entropy: I suggest to discuss the scaling of measuring OTOC and stabilizer entropy more in depth given recent results for both experiment ([47]) and numerics (arXiv:2209.10541), and do a proper comparison of complexity and experimental protocols. This discussion should give evidence whether OTOC is indeed better in some instances for measuring nonstabilizerness. We appreciate the reviewer's point on adding discussion on the recent papers. We included a comprehensive discussion about the similarities and the potential relationship between OTOC fluctuations and Tsallis stabilizer entropies on section IV paragraph 4.

  • "Interestingly, in the context of quantum error correction, the Clifford gates can be implemented fault-tolerantly": Note that this is not true for all error correction codes, some codes can implement non-Clifford gates transversally, but not all Cliffords. We thank the reviewer for pointing out this important point. We adjusted the language accordingly.

  • page 3: "Another magic monotone introduced for qubits is the Stabilizer Renyi Entropy": note that stabilizer entropy not a monotone (i.e. it is not defined for mixed states, and also not a monotone for pure states as channels that map pure to pure states for n<2 it has been shown not to be a monotone (arXiv:2303.10152)) We thank the reviewer for pointing out this important point. We changed the phrasing.

  • page 5: Why does the Ising model evolution is surrounded by Clifford gates? Why is this necessary? This should be explained more clearly. We thank the reviewer for raising this point. For our method to work, scrambling is an essential feature. Clifford blocks fulfil this role in the case of a random circuit. To the best of our understanding, the method we present in the manuscript requires such a low number of samples because we measure the circuit in the regime where the initial state is fully scrambled. Hamiltonian evolution alone does not necessarily have this feature and without the scrambling we do not have enough diversity in measured samples to be able to sample only a small subset of instances.

We added more detailed explanation in paragraph one of section III.C.

  • it is noted that stabilizer entropy can only be computed up to 12 qubits, however computations up to 15 qubits have been performed (arXiv:2209.10541). Thus, I am not sure the claim that computing OTOC is easier than stabilizer entropy is valid. We thank the reviewer for mentioning this reference and we apologise for overlooking it in our literature search. We adjusted our claims accordingly and added the reference.

  • It is mentioned in the conclusion that OTOCs are better to compute nonstabilizerness due to scalability. OTOCs can be measured efficiently by sampling, however similarly it has been noted that stabilizer entropy can be measured efficiently as well by sampling (e.g. [47]). In what sense is thus OTOC better than stabilizer entropy?

We agree with the referee’s point. In the present manuscript we can only show analytically the logarithmic relation to 2-Renyi entropy. At the same time, we provide strong numerical evidence that, at least for a classically simulatable number of qubit, the number of OTOC samples we need to exactly approximate Stabilizer Renyi Entropy and Mana is extremely low. For instance, for 12 qubits, where we would have to average 4^12 OTOC to satisfy the analytical formula, we obtain correct scaling and negligible numerical error with as little as 50 OTOC samples (see Figure 2 and 3). We admittedly do not provide analytical proof for this behaviour, but after thorough numerical testing, we wish to highlight the property that in practice, when sampling from a scrambled state, significantly lower number of samples is needed compared to what analytical relationship would suggest. We reformulated our conclusions accordingly.

  • references are not properly formatted and missing important info, e.g. 2, 8, 31, 47, 48 We thank the referee for pointing out the formatting issue. We corrected this.

  • I am wondering whether the equivalence of OTOC and nonstabilizerness could be made analytical, e.g. exact equivalence was found for the inverse participation ratio recently (e.g. see arXiv:2305.11797) We again acknowledge contribution of Dr. Lorenzo Leone during this peer-review process. We added the analytical relationship into the manuscript and provided detailed proof in Appendix C.

Anonymous Report 2

Strengths

-An interesting connection between magic monotones and quantum information scrambling is found. This link may have a practical importance in future. -A GitLab page containing the code used for the numerical simulations is provided

Weaknesses -The numerics should be improved, for instance addressing the scaling of required samples with the system size N -Some of the statement and messages are not very clear

Report The paper should be accepted by SciPost after some minor revisions and a general improvement of the clarity of the work (see the paragraph Requested changes).

Requested changes Questions/remarks/observations: -Regarding your sentence "[Mana is] defined for odd-dimensional local Hilbert space and thus not possible to evaluate for qubits": from a technical/mathematical point of view, what does make mana not well-defined for an even dimensional Hilbert spaces? For clarity, I think you should comment on this in the text.

We thank the reviewer for highlighting this. We commented on why mana is not calculable for qubits in section III. B paragraph one.

-Regarding Figure 2, how did you compute exactly the vertical error bars? In the caption, you write The vertical error bar is the statistical error calculated as the inverse of the square root of the number of samples". However, I guess that the statistical error associated with the standard deviation of OTOC should be related to the fourth momentum. Am I wrong? Regarding the horizontal error bars: did you include here the factor 1/√Nsamples coming from the fact you are estimating the fluctuations of the mean? Similar questions for the error bars in Figure 3.

We thank the referee for pointing this out. Previously, our error bar was precisely the statistical error of 1/sqrt(N), where N is total number of samples. This statistical behaviour dominates the error. Prompted by referee’s question we adjusted the error bar by repeating δotoc estimation 10 times and considered the std/sqrt( 10) of these different instances, thus including both standard deviation from the mean and statistical error. As seen in Figures 2 and 3, this new estimation decreased overall error bar significantly. We thank the referee for the suggestion.

-Do you think your analysis would be affected by considering other operators for the OTOC? Maybe you can add a comment about this. Also, for which reason you considered operators localized on the boundaries (XN−1 and Z1)? [By the way, if the chain has length N, I think the last site should be labelled by N...]

As long as the two operators, in our paper A and B, are commuting, these results hold, meaning we could apply these (or other Paulis) on any qubit pair in our circuit. We clarified our setup in Section II. B paragraph one.

The question about different types of correlators is very interesting. Here, we considered OTOC due to their interesting analytical relationship with the magic monotones we are attempting to approximate (proof added in revision, section IIIB). At the same time, we see that numerically, we can significantly undersample the analytical relationship of these two quantities and still obtain a very precise approximation. Designing other correlators that would have this property goes beyond the scope of what we did here, but it is nonetheless very stimulating question that might be addressed in the future research.

-I do not understand well the purpose of Figure 3c): are you claiming that you can estimate the second Stabilizer Renyi Entropy (SRE) with reasonable accuracy by increasing the number of sampled OTOC? In this case, I think you should perform a better analysis with increasing values of the number of samples.Also, why are you now forgetting about your linear fit coefficients and just estimating the average SRE as 1− δOTOC? And why you consider the observation that the absolute error for both sample counts is higher in the intermediate region of the T-gate count" as notable? We fully agree with the referee. In the new version of the manuscript we omitted this figure and added the analytical results instead.

-I think your statement that "OTOC fluctuations is the numerically lesser demanding method to estimate magic" should be made more precise. In particular, you have to address the scaling of the number of samples needed to reach a given accuracy with the system size N. The referee is completely correct at pointing out this challenge and it in fact highlights the central challenge in the field of non-stablizerness. Calculating non-stabilizerness measures is extremely numerically costly both for exact and for approximated quantum states, see Refs. [19, 55]. Our algorithm, with it’s beneficial scaling in number of OTOC instances is attempting to solve this challenge, but at the moment we are not yet able to show large system scaling beyond 12 qubits. Interesting question attached to this challenge is that the approximative methods do not necessarily maintain entanglement or non-stabilizerness structure of the state and this is a key challenge with deploying them. We are working actively towards scaling that is at the moment unfortunately beyond the scope of this manuscript.

-What is the main message of Figure 5 and in general of Section C? Indeed, you are not finding a perfect agreement between OTOC fluctuations of the circuit in Figure 5a) and the SRE of the time evolved state, just a similar behaviour... and so? Are you claiming that OTOC fluctuations are magic monotone as well?

We thank the referee for highlighting this point. In the first part of the manuscript we analyze the behavior of our method for discretized t-doped circuit setting. We were interested to see if it’s possible to formulate corresponding statements about OTOC sampling as a magic estimation for continuous Hamiltonian evolution as well. Indeed we find that with just a handful OTOC samples we can estimate magic generated by continuous time-evolution if less precisely. Note that we added the stabilizer blocks to achieve the scrambling that allows us to sample less. So indeed, we are claiming that averaging over low number of OTOC samples can play a role of magic monotone even in the continuous time evolution setting.

We reformulated paragraph 1 of section III.C to reflect this explanation. We thank the referee for pointing out the lack of clarity.

Other proposed changes: -Am I wrong or the letter q to indicate the local Hilbert space dimension is defined \textit{after} using it? I think it should be better to introduce the notation before of using it. We thank the referee for catching this mistake, we explained the notation in the first paragraph of section III.A.

-I think that when you report the results of a (linear) fit you should not use the symbol =, because it is not an exact mathematical equality. You can use ≃ instead. We thank the referee, we changed the notation.

-I think a reader would benefit from having the following additional references: https://arxiv.org/abs/2303.05536 and https://arxiv.org/abs/2304.01175. The first paper introduces a new method to evaluate efficiently the SRE for a Matrix Product State, by performing a Pauli sampling. I think it should be cited around the sentence "approximate numerical methods such as tensor networks". The second one regards a way of connecting the SRE with the properties of the entanglement spectrum, and I think can be considered as a general reference about SR

We thank to the referee for pointing out missing literature. We reviewed these references and added them to the manuscript.

Report 3 by Lorenzo Leone on 2023-9-7 Strengths 1) The paper addresses the highly challenging task of establishing a connection between magic and information scrambling. While there had been hints suggesting this connection, a systematic study like the one presented in this paper was absent. 2) The author's decision to investigate scrambling fluctuations is indeed well-suited for this purpose. Magic is not related to scrambling per se, being Clifford unitaries vey good scramblers of information. However, it extends beyond mere scrambling and delves into the intricacies of scrambling complexity, effectively captured by ensemble fluctuations (with respect to the Clifford average, as pointed out in the author's study). This complexity of scrambling plays a crucial role in determining whether the ensemble of unitaries possesses universal properties or not, and ultimately, it manifests in finer-grained aspects of scrambling, such as those exemplified by the 8-point OTOCs. Weaknesses

The paper's weakness lies in the lack of detailed numerical analysis, which remains incomplete. Given that the authors assert that measuring 4-OTOC fluctuations can offer a more efficient means of measuring magic compared to previous works, the scaling with N is of paramount significance to give sense this claim and, most importantly, sentences like '... for the system sizes studied in this work, we have not encountered any evidence that the sampling complexity should be exponential in the number of qubits...' to support the claim of the paper are better to be avoided. However, we are aware that stabilizer entropy can be computed numerically (for entangled states) up to N=12 and thus a finite size scaling becomes challenging. Report I recommend the paper for publication, but it is crucial that the authors undertake a significant revision. Although the topic holds great importance, and the chosen figure of merit is appropriate, there is substantial work that needs to be completed before the paper can be deemed ready for publication.

Requested changes

(A) As demonstrated in the attached PDF, the proposed proportionality δotoc=AM2+B between the 4-OTOC fluctuations δotoc (with respect to the Clifford average) and the second stabilizer entropy M2 is incorrect. Instead, despite OTOC fluctuations exhibit a positive correlation with the 2-stabilizer entropy, this relationship follows an exponential dependence upon the 2-stabilizer entropy. While the attached PDF provides analytical proof for this correction, there were already some indicators within the authors' work that hinted at this being the case:

We thank the referee for sharing this proof with us and significantly improving our manuscript. We re-fitted the proven analytical relation to our numerical results and indeed it offers much better agreement than our earlier fit.

1) Fig.(3)c illustrates that OTOC fluctuations and stabilizer entropy exhibit the most significant differences in the middle range, precisely where a linear line and a saturating exponential curve differ the most, as depicted in Fig. (1) of the attached PDF. Indeed in the light of the analytical proof we omitted Fig. 3C.

2) I believe that the log-plot of δotoc (and not 1−δotoc) in Fig. (3)b would result in the blue and red points aligning along a linear trend. This is indeed correct and implemented in the new version of the manuscript.

(B) The paper requires further numerical or analytical investigation to assess the efficiency of measuring magic through OTOC fluctuations. Otherwise, even if the authors are unable to present a finite size scaling that supports their assertion, the link between magic and OTOCs remains intriguing. However, in the latter case, all assertions regarding the efficiency of measuring magic through scrambling would be better to be omitted, including a potential alteration of the paper's title. It is essential to ensure that we do not inadvertently misguide the scientific community. Furthermore, in light of point (A), one can still measure the 2-stabilizer entropy via the OTOC fluctuations, although the functional dependence differs from the one suggested in the paper. It would be better to consider this in a future revision.

We thank the reviewer for directing us to this issue. Due to the limits to the exact simulatability of large system sizes, we agree that we cannot support the idea of efficiency by exact simulation. Our present algorithm does, however, show beneficial scaling in the number of OTOC samples compared to the analytical formulation. We highlighted this possible sampling advantage in a revised version without making statements of efficiency.

(C) I believe that a discussion on Eq.~(19) of (PhysRevA.106.062434) should be added as it is linking the complexity of scrambling, the 8-OTOC and the 4-OTOC fluctuations explored by the authors.

We added the discussion of this reference in the manuscript, specifically in the last paragraph of section III. B.

---

## Round 4 · List of Changes

- changing dependency of 2-Renyi entropy with δotoc
- In section IV paragraph 4, we added the discussion on the potential relationship between OTOC fluctuations and Tsallis stabilizer entropies
- adjusted the language about the Clifford gate and fault-tolerant computations.
- changing the phrase that stabilizer Renyi entropies are monotone
- more detailed explanation in paragraph one of section III.C about Clifford scramblers of the Hamiltonian time evolution case.
- fixed the format of references 2, 8, 31, 47, 48
- Adding analytical proof to Appendix C
- commented on why mana is not calculable for qubits in section III. B paragraph one
- changed the format of error bars in Figures 2 and 3
-clarified our setup on operators A and B in Section II. B paragraph one
-we omitted Fig. 3C.
-We highlighted this possible sampling advantage in a revised version without making statements of efficiency.

---

## Round 5 · List of Changes

Figure 4 updated

---

## Editorial Decision

published